# Reduction of Lams-Related Adverse Events with Accumulating Experience in a Large-Volume Tertiary Referral Center

**DOI:** 10.3390/jcm12031037

**Published:** 2023-01-29

**Authors:** Sebastian Stefanovic, Helena Degroote, Pieter Hindryckx

**Affiliations:** 1Department of Gastroenterology, University Hospital of Ghent, Corneel Heymanslaan 10, 1K12-IE, 9000 Ghent, Belgium; 2Diagnostic Center Bled, Pod Skalo 4, 4260 Bled, Slovenia

**Keywords:** EUS, LAMS, interventional EUS, adverse events, guideline

## Abstract

Background and aims: Lumen-apposing metal stents (LAMSs) are increasingly used both for on- and off-label indications. We continuously adapt our step-by-step protocol to optimize the safe deployment of LAMSs for the different indications. The aim of this study was to evaluate the impact of this approach over time. Methods: We conducted a single-center study on consecutive patients who underwent LAMS placement for on- and off-label indications between June 2020 and June 2022. Endpoints included technical success, clinical success and adverse event rates. We compared the results with our previously published early experience with LAMSs (N = 61), between March 2018 and May 2020. Results: This cohort consisted of 168 LAMSs in 153 patients. Almost half of them (47.6%) were placed for off-label indications (gastro-enterostomy, temporary access to the excluded stomach in patients with previous gastric bypass, drainage of postsurgical collections, stenting of short refractory gastrointestinal strictures). While the technical and clinical success rates were similar to those in our previously published cohort (97% and 93.5% versus 93.4% and 88.5%, respectively), the adverse event rate dropped from 21.3% to 8.9%. Conclusions: Our results demonstrate the impact of a learning curve in LAMS placement, with a clinically relevant drop in LAMS-related adverse events over time.

## 1. Introduction

Lumen-apposing metal stents (LAMSs) were first approved by the FDA in 2013 for drainage of peripancreatic collections [1,2,3], but their use quickly expanded to manage complicated situations that were previously referred to interventional radiology and/or surgery.

Currently, the use of LAMSs is also approved for gallbladder drainage in nonsurgical candidates and bile duct drainage in cases of failed endoscopic retrograde cholangiopancreatography (ERCP) and/or malignant distal biliary obstruction. Off-label indications include the creation of a luminal anastomosis (e.g., to alleviate gastric outlet obstruction in cases of duodenal obstruction or to create temporary access to the excluded stomach for endoscopy in gastric bypass patients), the drainage of postsurgical collections and the management of short refractory gastrointestinal strictures [3,4].

Although the clinical benefit of LAMSs may be substantial for many patients, one should be aware of potential (serious) adverse events related to LAMS procedures. We have previously reported an adverse event of 21.3%. Choi et al. [5] recently published similar results in the largest cohort study to date.

As for any other endoscopic intervention, three criteria should be fulfilled to minimize the adverse event rate and related morbidity: First, one should be fully aware of the potential complications of the procedure (what can happen?). Second, one should know the strategies to maximally prevent these complications (how to prevent it from happening?). Finally, one should be able to manage the complications in the appropriate way (do I have a plan B if it happens?).

The recent publication of expert consensus guidelines for interventional endoscopic ultrasound is very helpful in this regard, but data on young and evolving endoscopic techniques, such as LAMS placement, remain scarce and make large cohort studies of particular value.

The aim of our study was to assess the impact of a learning curve in LAMS placement in terms of technical success, clinical success and adverse event rate.

## 2. Methods

### 2.1. Study Design/Population

This was a retrospective single-center cohort study of consecutive patients who underwent LAMS placement (Hot AXIOS stent, Boston Scientific Corporation, Natick, MA, USA) at our tertiary referral center between June 2020 and June 2022. The study was reviewed and approved by the institutional ethical review board (reference: ONZ-2022-0179).

Patients were divided into categories: A: drainage of peripancreatic collections, B: biliary drainage (CBD), C: gallbladder drainage (GBD), D: gastroenteric anastomosis, E: temporary gastric access for endoscopy (GATE), F: treatment of refractory gastrointestinal (GI) strictures and G: miscellaneous, other indications.

### 2.2. Data Collection and Analysis

Using the electronic medical records and our prospectively collected database (for internal quality monitoring), we collected data on patient demographics, indications, technical and clinical success rates and adverse events of all LAMS procedures performed at our department during the period of interest.

Technical success was defined as the successful deployment of the LAMS in the desired position.

Clinical success was defined based on a previously published manuscript regarding the use of LAMSs [6]. For refractory anastomotic strictures, we used the following definition: normal oral intake for the anticipated period of stenting (12 weeks).

The severity of adverse events was graded based on the newly published American Society for Gastrointestinal Endoscopy AGREE classification [7]. Adverse events were recorded based on retrospective EMR reviews.

All obtained results were compared with those from our previously published cohort [6]. This historical cohort included all consecutive cases since the introduction of LAMSs in the University Hospital of Ghent, Belgium, all performed by PH who had no previous experience with LAMSs (except for training in models). The current cohort includes all consecutive patients after this historical cohort. All cases were performed by PH or HDG in direct supervision of PH. Data analysis was performed with SPSS 25 statistical software (IBM, Chicago, IL, USA). Proportions were compared using the chi-square test for 2 × 2 tables. A 2-sided *p* value < 0.5 was considered statistically significant.

## 3. Results

### 3.1. Indications

We included 168 procedures performed in 153 patients. Fifteen patients had received more than one LAMS placement for different indications. The patient characteristics are provided in Table 1.

Eighty-eight procedures (52.4%) were performed for on-label indications, and seventy-eight (47.6%) were performed for off-label indications. The numbers for each indication can be found in Table 2. A comparison with the previously published cohort can be found in Appendix A.

### 3.2. Technical Success

The technical success rate in our current cohort was higher (163/168; 97%; Table 3) but not significantly different from our historical cohort (57/61; 93.4%; *p* = 0.22). Procedure outcomes of LAMS placement for the different on- and off-label indications can be found in Table 2.

In five patients, LAMS deployment was either not possible (too long a distance between the GI lumen and the target, N = 2, both in the EUS-GE group) or considered unsafe (due to poor visualization, N = 2, both in the PFC group). One misplacement occurred in a patient with known metastatic rectal carcinoma who needed drainage of a pararectal abscess.

### 3.3. Clinical Success Rate

The clinical success rate in our current cohort was higher (157/168; 93.5%, Table 3) but not significantly different from our historical cohort (54/61; 88.5%; *p* = 0.21).

Clinical failures included the five patients with technical failures described above. The remaining six cases included three patients with peripancreatic fluid collections, one patient with malignant distal biliary obstruction and two patients with refractory esophageal strictures.

### 3.4. Adverse Events

The adverse event rate was significantly lower (15/168; 8.9%, Table 2) than that in our historical cohort (13/56; 21.3%; *p* = 0.01) (Table 4). We performed a comparative subanalysis between the current and the historical cohort for the two most frequent LAMS indications (PFC and CBD), clearly demonstrating a caseload-dependent reduction in adverse events (Figure 1A,B). All AEs were categorized based on the AGREE criteria and are described below [7].

#### 3.4.1. Grade II AEs

One grade II adverse event was seen in the choledochoduodenostomy (CDS) (Group D). The patient showed symptoms of postprocedural gastrointestinal bleeding with melena. However, despite using imaging diagnostics, we were unable to identify the cause of the bleeding (EGD, colonoscopy and CTA). The patient’s progress was uneventful following conservative treatment and fluid resuscitation.

#### 3.4.2. Grade III AEs

A total of 15 IIIa adverse events occurred. Five patients (two in the walled-off necrosis (WON) group, two in the CDS group and one in the EUS-GE group) had postprocedural gastrointestinal bleeding needing endoscopic intervention. One patient in the CDS group required a blood transfusion in addition to endoscopic care.

The LAMS had to be removed or endoscopically replaced in eight patients either due to tumor overgrowth (N = 2), enterocolonic fistula (N = 1), air and fluid leaks (N = 1), stent migration (N = 3) or ascending cholangitis (N = 1).

One of the most serious adverse events was observed in a patient who had a rectal abscess from metastatic rectal malignancy. A subsequent CT scan revealed that the distal flange had been deployed into the muscle tissue as a result of low visibility. The LAMS was removed, but the fistula was not closed because of poor bowel preparation. Due to the patient’s complex surgical history and his poor prognosis, conservative management was initiated after multidisciplinary discussion. Although the patient did not die from this adverse event, postprocedural chronic pain persisted until his death.

#### 3.4.3. Grade IIIb AEs

One patient experienced a grade IIIb AE that necessitated surgery. The patient with a history of gastric bypass underwent LAMS gastro-gastrostomy to perform ERCP (common bile duct stone). Although the interval between LAMS placement and ERCP was only 8 days, LAMS dislocation during ERCP led to a perforation. The defect was successfully closed from the stomach pouch with an over-the-scope, but the patient developed signs of peritonitis, necessitating laparoscopic surgery with a smooth recovery afterward.

There were no grade I, IVa, IVb, or grade V AEs in our cohort.

## 4. Discussion

The results of our cohort clearly demonstrate the impact of a learning curve on the outcome of LAMS procedures. Over time, we observed only a slight (non-significant) increase in technical and clinical success rates but a significant and relevant drop in the adverse event rate.

Our general safety measures across all indications to prevent bleeding and misplacement of the LAMS include preprocedural cross-sectional imaging, the use of Doppler imaging, measurement of the distance between the gastrointestinal (GI) tract lumen and the target, taking time to find the best scope position, the immediate removal of the electrocautery cable once the target has been penetrated with the catheter and the use of the appropriate LAMS diameter.

The use of LAMSs was first authorized for peripancreatic fluid collections. Most of the data regarding the use of LAMSs come from this indication [1,8,9,10,11,12,13,14]. Our technical (95.0%) and clinical (87.5%) success rates were in line with recently published data [8,9].

We observed a notable drop in LAMS-related adverse events for this indication, from 33% in our previous cohort to 5% in the current cohort. Similar to our observations, Facciorusso et al. [10] found that adverse events decrease with increasing caseload. Over time, we took the following measures to achieve this low adverse event rate: a systematic use of periprocedural antibiotics, the use of broad diameter stents (preferably 20 mm) in case of WON, the use of double pigtail catheters inside the LAMS lumen to protect patency by preventing complete blockage of the lumen by food or necrotic tissue and early follow-up imaging (after 1–2 weeks in case of pseudocysts and 3–4 weeks in case of WON) to check for resolution of the collection and to ensure timely removal of the LAMS with or without replacement by pigtail catheters. The latter is important to avoid severe bleeding resulting from erosion of the PFC wall by the LAMS and in line with previous recommendations [9,11,12,13,14,15]. A recent study published by Najar et al., however, found no increase in adverse events if the LAMS was removed after more than four weeks [16].

Choledochoduodenostomy is the second approved indication for LAMS in patients with failed ERCP and/or malignant distal biliary obstruction. The benefit over percutaneous biliary drainage (PTBD) in this indication has now been clearly demonstrated [17]. While ERCP remains the procedure of choice for the management of obstructive biliary drainage, a potential paradigm shift toward first-line choledochoduodenostomy has been claimed in the context of inoperable malignant distal biliary obstruction and is currently under further investigation in the ELEMENT trial [18].

One of the most important messages is that one should be trained in small-space LAMS placement before proceeding with this indication since the consequences of misplacement are severe [6]. We now use a guidewire for bile ducts < 14 mm (introduced after freehand introduction of the LAMS catheter into the bile duct) to allow for a rescue intervention (covered self-expanding metal stent (SEMS) placement) in case of misplacement.

LAMS misplacement and persistent or recurrent cholestasis (20%) was also a major issue in the Choi et al. [5] group. This was due to biliary food impaction. Our personal experience is that it can be avoided by always using a 6 × 8 mm LAMS size independent of the bile duct diameter. This issue of the unavailability of narrower stents in the USA and subsequent adverse events related to food impaction was also noted in the meta-analysis by Peng et al. [19] In this regard, one ongoing RCT investigates the added value of a pigtail inside the lumen of a small-to-medium size LAMS (BAMPI trial) [20].

LAMSs are approved for gallbladder drainage in nonoperative candidates and should be preferred over percutaneous drainage if expertise is available [21,22,23,24]. In our cohort, we noted a high technical and clinical success rate with low rates of AEs.

We believe that the transduodenal route should be preferred over the transgastric route since the risk of migration is much lower. Early migration of the LAMS (<1 week after placement) might lead to gastric perforation and biliary peritonitis that can only be resolved with surgery in a high-risk population. In addition, one should be aware of the risk of pyloric obstruction by the proximal flange of the LAMS if the gastric route is chosen. Finally, as in our own experience, future cholecystectomy is not hampered by the presence of a cholecystoduodenostomy.

In the absence of gallbladder stones, we place an 8/8 or 10/10 stent, sufficient for adequate drainage in our series. A larger diameter LAMS (15/10) is preferred in cases with stones present. We recommend a double pigtail inside the lumen of the LAMS, especially in a case with stones, to preserve the LAMS patency.

EUS-guided gastroenterostomy (EUS-GE) to alleviate gastric outlet obstruction is a promising but off-label indication for LAMSs. Retrospective studies and our personal experience suggest that EUS-GE is safe, provides better functional results than duodenal stenting and is associated with a quicker recovery and a shorter hospital stay compared to surgery [25,26,27]. While different techniques of EUS-GE exist [27,28,29,30], we only use the direct anterograde method. Technical and clinical success rates were high (94.3%), which is in agreement with the available data [28,29].

Stent maldeployment might have serious consequences in cases of a jejunal perforation that, in most cases, will not be accessible for endoscopic closure. Accidental deployment in the colon instead of the jejunum is another potential hazard. Careful selection of the best position, adequate distension of the jejunal loop with colored saline, a test puncture in case the jejunal filling catheter cannot be seen with EUS, the use of antispasmodics and short-term apnea are all helpful to minimize the risk of maldeployment.

To maximize the functional results of EUS-GE, we always use a 20 mm LAMS for this indication. We dilate the lumen up to 18 mm after deployment and clipping of the LAMS to hasten the time to normal oral dietary intake, but further data to support this approach are lacking [30].

According to Jovani et al., 25 procedures can be considered as the threshold to achieve proficiency in EUS-GE [31].

Twenty-five patients in our cohort had LAMSs placed for temporary endoscopic access (GATE) due to the need for other endoscopic procedures (ERCP). We noted 100% technical and clinical success rates. These results are better than those available in the literature [32,33].

Based on one serious adverse event in our institution and similar to Wang et al. [32], we now use a 20 mm LAMS if possible and delay ERCP for at least 2 weeks after placement if possible. If there is a need for emergency ERCP, we fix the LAMS with an over-the-scope stent fix clip (OTSC) and leave a guidewire in the excluded stomach upon withdrawal of the duodenoscope to allow for a rescue procedure (covered stent placement) in case of dislocation. We have only performed three of these one-step cases, all without complications.

We sometimes use LAMSs for refractory anastomotic strictures to minimize the risk of stent migration and stent-related inflammation. However, our experience is too limited to provide any recommendations for this indication. The potential use of LAMSs for short gastrointestinal strictures has previously been proposed by others [34].

Our study has some limitations. It was performed in a single high-volume tertiary academic referral center. All procedures were performed by two expert endoscopists with extensive experience in EUS and ERCP (HDG and PH). Outcomes may vary according to local expertise. One other limitation of the study was the use of only one LAMS type (Hot AXIOS). Our results might not be fully applicable to other LAMSs on the market. Although our study was retrospective, the data were carefully recorded, and there are no missing data.

In summary, our study demonstrates the most important impact of a learning curve in LAMS placement is a significant drop in complications over time due to protocol optimization. Expert consensus guidelines on the safe deployment of LAMSs for the different indications are crucial to validate our recommendations and reduce the overall risk of LAMS-related adverse events in the broader community.

## Figures and Tables

**Figure 1 jcm-12-01037-f001:**
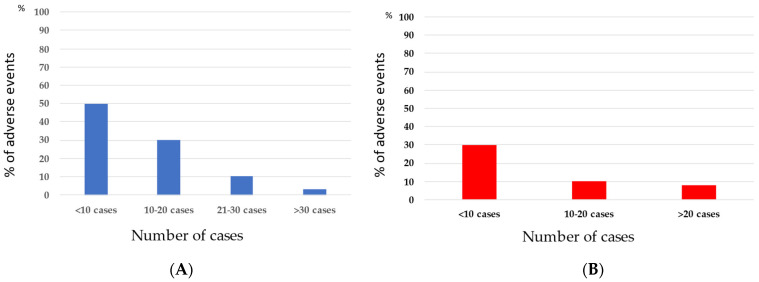
Caseload-dependent reduction in adverse events in EUS drainage of PFCs and CBD. Caseload-dependent reduction in AEs in EUS drainage of PFCs (**A**) and bile duct drainage (**B**) based on a combined subanalysis of the current cohort and a previously published historical cohort [6]. AE—adverse events, EUS—endoscopic ultrasound, PFC—peripancreatic fluid collection, CBD—common bile duct.

**Table 1 jcm-12-01037-t001:** Baseline characteristics of patients.

Characteristic	Values *n*, (%)
**No. of patients**	153
**No. of procedures**	168
**Median age, y, IQR**	62.8 [49.9–72.9]
**Sex, M: F**	93 (57.4): 69 (42.6)
**ASA score**	II 96 (59.3) III 66 (40.7)
**ECOG PS score**	I 22 (13.6) II 88 (54.3) III 51 (31.5) IV 1 (0.6)

IQR—interquartile range, ASA—American Society of Anesthesiology Score, ECOG PS—performance status.

**Table 2 jcm-12-01037-t002:** Procedural details of patients undergoing LAMS placement.

Indication (N = 168)	Number of Procedures	Location	Technical Success	Clinical Success	Adverse Events	Underlying Malignancy
**PFC** - **Pseudocyst** - **WON**	N = 40 7 (17.5) 33 (82.5)		38 (95.0) 7 (100) 31 (93.9)	35 (87.5) 7 (100) 28 (84.8)	2 (5.0) 0 (0.0) 2 (6.1)	0 (0) 0 (0) 0 (0)
**GE** - **Benign GOO** - **Malignant GOO**	N = 35 11 (31.4) 24 (68.6)	GG = 4 (11.4) GJ = 31 (88.6)	33 (94.3) 11 (100) 22 (91.7)	33 (94.3) 11 (100) 22 (91.7)	3 (8.6) 0 (0.0) 3 (12.5)	24 (68.6) 0 (0.0) 24 (100)
**EUS-BD**	N = 21		21 (100)	20 (95.2)	2 (9.5)	17 (81.0)
**EUS-GBD**	N = 27		27 (100)	27 (100)	5 (18.5)	17 (63.0)
**GATE**	N = 25		25 (100)	25 (100)	1 (4.0)	1 (4.0)
**Treatment of refractory GI strictures**	N = 6	Esophagus 5 (83.3) Pyloric Channel 1 (16.7)	6 (100)	4 (66.7)	0 (0)	0 (0)
**Miscellaneous**	N = 12		11 (91.7)	11 (91.7)	3 (25.0)	1 (8.3)

PFC—peripancreatic fluid collections, GE—gastroenterostomy, EUS-BD—endoscopic ultrasound guided biliary drainage, EUS-GBD—endoscopic ultrasound guided gallbladder drainage, GATE—temporary access for endoscopic procedures, GI—gastrointestinal.

**Table 3 jcm-12-01037-t003:** Outcomes overall. Graded according to the AGREE classification [7].

Characteristic, N = 168	Values N, (%)
Technical success	163/168 (97.0)
Clinical success	157/168 (93.5)
Adverse events	17 (10.1)
1	0 (0)
2	1 (0.1)
3a	15 (8.9)
3b	1 (0.6)
4a	0 (0)
4b	0 (0)
5	0 (0)

**Table 4 jcm-12-01037-t004:** Complications, management and outcome for respective LAMS indications.

Indication	Adverse Event Based on AGREE [7]	Frequency	Description of Event	Management	Outcome
**PFC**	1 2 3a 3b 4a 4b 5	0 0 2 0 0 0 0	Bleeding (N = 2)	Endoscopy (N = 2)	Resolved (N = 2)
**GE**	1 2 3a 3b 4a 4b 5	0 0 3 0 0 0 0	Colono-enteral fistula (N = 1) Ulcers at the level of jejunum (N = 1) Bleeding (N = 1)	Endoscopy (N = 3)	Resolved (N = 3)
**CBD**	1 2 3a 3b 4a 4b 5	0 0 4 0 0 0 0	Overgrowth of AXIOS with tumor tissue (N = 1) Bleeding requiring bicap and transfusion (N = 1) Ascending cholangitis (N = 1) Uncertainty about position of distal flange (N = 1)	Endoscopy (N = 4)	Resolved (N = 4)
**GBD**	1 2 3a 3b 4a 4b 5	0 1 1 0 0 0 0	Bleeding (N = 2)	Medical treatment (N = 1) Endoscopy (N = 1)	Resolved (N = 2)
**GATE**	1 2 3a 3b 4a 4b 5	0 0 0 1 0 0 0	Dislocation of LAMS (N = 1)	Surgery (N = 1)	Resolved (N = 1)
**Miscellaneous**	1 2 3a 3b 4a 4b 5	0 0 3 0 0 0 0	Aberrant LAMS position (N = 1) Leakage of collection fluid into peritoneum (N = 1) LAMS migration (N = 1)	Medical treatment (N = 2) Endoscopy (N = 1)	Resolved (N = 1) Non-resolved (N = 1)

LAMS—lumen apposing metal stent, PFC—peripancreatic fluid collections, GE—gastroenterostomy, EUS BD—endoscopic ultrasound guided biliary drainage, EUS GBD—endoscopic ultrasound guided gallbladder drainage, GATE—temporary access for endoscopic procedures, GI—gastrointestinal.

## Data Availability

Data supporting reported results was generated with the use of electronic medical records. The data presented in this study are available on request from the corresponding author.

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
