# Peer review of "Reduction of Lams-Related Adverse Events with Accumulating Experience in a Large-Volume Tertiary Referral Center"

_jcm, 2023, doi:10.3390/jcm12031037_

Round 1
Reviewer 1 Report
This is a retrospective large paper investigating an important point, such as the learning curve for the deployment of lumen apposing metal stent (LAMS). The LAMS used was the Hot Axios and patients were divided according to the indication. Technical success, clinical success and adverse events were evaluated and compared with an historical cohort.
Here are listed my suggestions:
- You should specify the experience of the operator(s) at the time of this cohort and of the historical cohort. Moreover, some more data about that historical cohort should be provided (even if it was previously published). For example, were the number of different indications comparable? I suggest to provide a baseline table comparing the two cohorts.
- Line 186: "IVb" should be IIIb, I guess.
- In table 3 add "severity" after adverse events.
- There are no data about the indwelling time of LAMS that is a debated point that can be associated with AEs.
- There are some papers on this topic that should be mentioned and discussed (PMID: 35217579; PMID: 34637805; PMID: 34932991; PMID: 34331485; PMID: 34339667; PMID: 32066622)
Author Response
REVIEWER 1:
This is a retrospective large paper investigating an important point, such as the learning curve for the deployment of lumen apposing metal stent (LAMS). The LAMS used was the Hot Axios and patients were divided according to the indication. Technical success, clinical success and adverse events were evaluated and compared with an historical cohort.
Here are listed my suggestions:
- You should specify the experience of the operator(s) at the time of this cohort and of the historical cohort. Moreover, some more data about that historical cohort should be provided (even if it was previously published). For example, were the number of different indications comparable? I suggest to provide a baseline table comparing the two cohorts.
Reply: We thank the reviewer for their comments. The first cohort (previously published, Hindryckx et al. Surg Endosc 2020) included all consecutive cases since the introduction of LAMS in the University Hospital of Ghent, Belgium. All cases were performed by PH who had no previous experience (except for training in models). The current cohort includes all consecutive patients after this historical cohort. All cases were performed by PH or HDG in direct supervision of PH.
Changes made to the manuscript: We have provided extra information in the “Data collection and analysis« section of the manuscript. To give more insight in the distribution of indications between the historical and the new cohort we have added a supplementary table to the manuscript. We refer to the table in the results section of our manuscript.
- Line 186: "IVb" should be IIIb, I guess.
Reply: We thank the reviewer for noting this error. We corrected it.
- In table 3 add "severity" after adverse events.
Reply: The adverse events in Table 3. are graded with numbers accordingly to the new ASGE Agree consensus. The exact definitions are provided in the paper that is referenced in our manuscript.
Changes made to the manuscript: We have added an asterisk in Table 3, with a reference to the AGREE consensus paper.
- There are no data about the indwelling time of LAMS that is a debated point that can be associated with AEs. - There are some papers on this topic that should be mentioned and discussed (PMID: 35217579; PMID: 34637805; PMID: 34932991; PMID: 34331485; PMID: 34339667; PMID: 32066622)
Reply: The reviewer raises an important point. For peripancreatic collections, regression of the collection after LAMS placement should be monitored to allow for timely removal of the LAMS in order to prevent erosion of the collection wall by the LAMS. The latter may result in important bleeding. We perform cross-sectional imaging after 2 weeks and 4 weeks for pseudocysts and walled-off necrosis, respectively. This is also covered in the discussion of our manuscript. The risk of long-term indwelling of LAMS in other indications is less clear. After a couple of months, the LAMS may lose coverage which will hamper extraction or can lead to a buried LAMS. In non-palliative patients, we therefore tend to avoid long-term indwelling of the LAMS. For example, in non-palliative patients that received a LAMS for gallbladder drainage, we tend to replace the LAMS by double pigtails after 4-6 weeks. The risk of patency loss of the conduit in case of a gastrojejunostomy seems to be low, even after long-term indwelling of the LAMS. We have several palliative patients with a perfectly patent LAMS gastrojejunostomy in situ after 2-3 years.
Changes made to the manuscript: We now explain the importance of timely removal of the LAMS in the discussion of the manuscript. We have added some of the suggested references.
Reviewer 2 Report
An interesting paper that examines the problems with LAMS in various situations. Well done and interesting study that looks at a specific question.
1. In the present study, for example, we believe that in EUS-BD and gastrointestinal stenosis, regular metal stents are also used in most cases, but we would like to know why you dared to use LAMS, so please describe it.
2. Although the title emphasizes the learning curve, the results do not mention it at all. We believe that the relationship between the learning curve and comorbidity must be examined.
Author Response
REVIEWER 2:
An interesting paper that examines the problems with LAMS in various situations. Well done and interesting study that looks at a specific question.
- In the present study, for example, we believe that in EUS-BD and gastrointestinal stenosis, regular metal stents are also used in most cases, but we would like to know why you dared to use LAMS, so please describe it.
Reply: Biliary drainage is an FDA- and EMEA-approved indication for LAMS. According to our experience, electrocautery-enhanced LAMS placement is technically less demanding than regular metal stent placement. The latter requires multiple material exchanges. With regard to gastrointestinal stenosis: we only consider LAMS for simple, short but refractory anastomotic strictures. The benefit of LAMS in this indication is a lower risk of stent migration and less chance of stent-related inflammation with subsequent stenosis. We all know the cases where regular esophageal stents for short strictures induced inflammation resulting in a second or longer stenosis as compared to the original one.
Changes made to the manuscript: We added the rationale for using LAMS in gastrointestinal strictures to the discussion part of our manuscript.
- Although the title emphasizes the learning curve, the results do not mention it at all. We believe that the relationship between the learning curve and comorbidity must be examined.
Reply: We thank the reviewer for this comment. The results of our cohort clearly demonstrate the impact of a learning curve on the outcome of LAMS procedures. Over time, we observed only a slight (nonsignificant) increase in technical and clinical success rates but a significant and relevant drop in the adverse event rate. To further elaborate on this matter, we performed a sub analysis for the two most frequent LAMS indications (PFC (Figure 1A.) and CBD (Figure 1B.), clearly demonstrating a caseload-dependent reduction in adverse events (Figure 1). The caseload for other indications was too low in the historical cohort to perform a meaningful sub analysis.
Changes made to the manuscript: Addition of figure 1 demonstrating the caseload-dependent reduction in adverse events for the two most frequent LAMS indications.
Round 2
Reviewer 1 Report
I have no further comments
Author Response
Thank you for the peer review of our manuscript
Reviewer 2 Report
1. As I said before, the title does not match the results at all. Likewise, the items under consideration and the conclusion do not match at all. If you are talking about a learning curve, you should be examining how the success rate changes with the number of cases experienced by young doctors, which is not the case here. At the very least, the title should be changed, the term "learning curve" should be removed, and it should be our hospital's experience with EUS-BD.
2. The number of doctors performing the procedure may have changed in the past, and what about the number of years of experience of those doctors? It is strange that even that is not mentioned.
3. This study only compares the success rate of EUS-BD and complications between the early and late groups, and the number of cases is not large, so the term "large single center experience" is an exaggeration.
Author Response
- As I said before, the title does not match the results at all. Likewise, the items under consideration and the conclusion do not match at all. If you are talking about a learning curve, you should be examining how the success rate changes with the number of cases experienced by young doctors, which is not the case here. At the very least, the title should be changed, the term "learning curve" should be removed, and it should be our hospital's experience with EUS-BD.
Reply: Apologies but we tend to disagree with this comment. We believe that our results clearly reflect the impact of a learning curve of PH in LAMS placement (from no experience in 2018 towards >250 cases in June 2022). We refer to our reply to the 2nd peer reviewer comment: all procedures in the historical cohort were performed by PH and almost all procedures in the new cohort were either performed or assisted by PH. For the two most common indications (PFC and BD) we could demonstrate a caseload-dependent reduction in adverse events (figure 1).
Changes made to the manuscript: We prefer to keep the title but if both the peer-reviewer and the editors feel that a change is needed we propose to modify the title into a less appealing: “Evolution of LAMS-related outcomes in a large-volume tertiary referral centre”
- The number of doctors performing the procedure may have changed in the past, and what about the number of years of experience of those doctors? It is strange that even that is not mentioned.
Reply: In the revised version of our manuscript, we clearly mention that the historical cohort included all consecutive cases since the introduction of LAMS in the University Hospital of Ghent, Belgium and were all performed by PH who had no previous experience with LAMS (except for training in models). The current cohort includes all consecutive patients after this historical cohort, performed by only two doctors (PH or HDG in direct supervision of PH). In other words, with the exception of 1-2 more easy cases performed by HDG during holidays of PH, PH performed or assisted all LAMS procedures. PH started with LAMS cases in March 2018. Data until June 2022 were included in the manuscript. At that time PH had reached 4 years and 3 months of experience in LAMS placement. The historical cohort represents the first 2 years, the new cohort the second 2 years.
- This study only compares the success rate of EUS-BD and complications between the early and late groups, and the number of cases is not large, so the term "large single center experience" is an exaggeration.
Reply: The total number of LAMS cases in our centre was 229 in June 2022. Currently we have about 100 LAMS cases/year. Based on feedback of national and international colleagues, this is considered a huge caseload. In figure 1 we provide the caseload-based reduction in adverse events for the two most common LAMS indications (PFC and BD), together accounting for 98 patients in our cohorts (64 PFC, 34 BD).
Changes made to the manuscript: We prefer to keep the title but if both the peer-reviewer and the editors feel that a change is needed we propose to modify the title into: “Evolution of LAMS-related outcomes in a large-volume tertiary referral centre”